# Rethinking Biosynthesis of Aclacinomycin A

**DOI:** 10.3390/molecules28062761

**Published:** 2023-03-18

**Authors:** Ziling Xu, Pingfang Tian

**Affiliations:** Beijing Key Laboratory of Bioprocess, College of Life Science and Technology, Beijing University of Chemical Technology, Beijing 100029, China

**Keywords:** aclacinomycin A, antitumor agent, biosynthesis, glycosyltransferase, metabolic engineering

## Abstract

Aclacinomycin A (ACM-A) is an anthracycline antitumor agent widely used in clinical practice. The current industrial production of ACM-A relies primarily on chemical synthesis and microbial fermentation. However, chemical synthesis involves multiple reactions which give rise to high production costs and environmental pollution. Microbial fermentation is a sustainable strategy, yet the current fermentation yield is too low to satisfy market demand. Hence, strain improvement is highly desirable, and tremendous endeavors have been made to decipher biosynthesis pathways and modify key enzymes. In this review, we comprehensively describe the reported biosynthesis pathways, key enzymes, and, especially, catalytic mechanisms. In addition, we come up with strategies to uncover unknown enzymes and improve the activities of rate-limiting enzymes. Overall, this review aims to provide valuable insights for complete biosynthesis of ACM-A.

## 1. Introduction

Cancer poses a severe threat to humans, and its resulting economic burden may increase in the future [1,2]. Although cancer incidence and fatality rates may recede due to early diagnosis, the treatment cost remains unaffordable for most patients [3]. To address this issue, researchers are gearing up to develop efficient antitumor medicines, and impressive advancements have been achieved in this regard. Extant antitumor medicines are roughly divided into eight categories, including alkylating agents [4], hormone antagonists [5], antimetabolites, cytotoxic antibiotics [6], plant alkaloids [7], platinum compounds [8], protein kinase inhibitors [9], and monoclonal antibodies [10]. Belonging to cytotoxic antibiotics, anthracycline antitumor agents have been widely used for the clinical treatment of cancer for half a century. Of the reported anthracycline antitumor agents, aclacinomycin (ACM), also known as aclarubicin (ACL), is actually the mixture of ACM-A, ACM-B, and ACM-Y, typically synthesized by *Streptomyces galilaeus* ATCC 31133 [11,12]. ACM-A manifests a long side-chain structure, in which three sugar residues, including L-rhodosamine, 2-deoxyfucose, and L-cinerulose A, are tethered to aklavinone. As an anticancer medicine used in Japan and China, ACM-A exhibits remarkable inhibitory activities against leukemias, gastric cancer, and lung cancer [13,14,15,16]. In principle, anthracyclines exhibit antitumor activity due to the targeting of the minor groove of DNA to form a stable complex [17] (Figure 1). This reversible targeting can repress topoisomerase II and therefore hamper the transcription and replication of DNA in this complex. Consequently, double-stranded DNA unwinds and the accumulated DNA fragments, triggering programmed cell death [18]. ACM-A was also found to evict histones and damage chromatin [19,20]. Compared to doxorubicin (DXR) and daunorubicin (DNR), the two anthracyclines isolated from *Streptomyces peucetius* ATCC 27952, ACM-A exhibits higher antitumor activity but much lower cardiotoxicity [21].

The chemical synthesis of ACM-A is challenging due to its bulky and complex structure. Aklavinone is an important member of tetrahydronaphthol whose benzylic hydroxy group can be easily eliminated. In previously reported synthesis, Diels–Alder reactions were found to be crucial for the synthesis of aklavinone, which raised the issue of regioselectivity [22,23]. It was reported that refinement led to only a 68% yield ratio of oligosaccharides in ACM-A [24,25]. Clearly, high stereoselectivity is desired, but the modulation of stereoselectivity is laborious and troublesome [26]. Although strategies have been developed (e.g., donor-controlled stereoselectivity), the multi-step syntheses of ACM-A is tedious and seems to be inevitable [27,28]. In addition, the organic synthesis of ACM-A shows shortcomings such as the need for toxic catalysts, low yield, and severe pollution [29]. In contrast, the bioproduction of ACM-A shows merits such as low cost, easy manipulation, and less environmental pollution. ACM-A is mainly obtained by the fermentation of *Streptomyces* sp.NO.MA144-MI ATCC 31133 or its mutants. Of reported *Sreptomyces* species, *Streptomyces lavendofoliae* is the most employed for the industrial production of ACM-A [30]. Since ACM-A is a secondary metabolite and indirectly produced from the core metabolic pathways in *Streptomyces*, its production is therefore low and needs to be substantially enhanced. Synthetic biology and metabolic engineering raise the hope for the high-level production of ACM-A based on rational design and the subsequent construction of biological systems. In this review, we not only outline the reported biosynthetic pathways of ACM-A, but also discuss the catalytic mechanisms of key enzymes. Furthermore, we come up with protocols for the overproduction of this economically important medicine.

## 2. Biosynthesis of ACM-A

The history of deciphering ACM-A biosynthesis dates back to the 1970s. In 1975, ACM-A and ACM-B were first isolated from the fermentation broth of *S. galilaeus* MA144-M1(ATCC 31133) [11]. In 1979, ACM-A was extracted using organic solvents. After undergoing silicic acid column chromatography, the extracts from the fermentation broth were condensed into crystal or microcrystalline powder [31]. In 1981, partial biosynthesis pathways of ACM-A were identified through the genetic loss of glycosylation and isotope labeling [32]. Afterwards, the polyketide synthase (PKS) genes from *S. galilaeus* ATCC 31615 were cloned and sequenced [33]. So far, a partial ACM-A pathway has been unraveled (Acc. NO. AF264025.1; Acc. NO. AF257324.2) (Figure 2). With one molecule of propionyl-CoA as a starter unit and nine molecules of malonyl-CoA as extender units, the type Ⅱ PKS catalyzes the formation of aklavinone through nine cycles of reactions, involving the incorporation of malonyl-CoA, cyclization, and oxidation. ACM-A is generated upon specific glycosylation and modification (Figure 3) [34,35,36]. ACM-A can be converted to ACM-Y by oxidoreductase [36]. The oxidoreductase located in cytoplasm catalyzes ACM-A to ACM-B, while the oxidoreductase in periplasm catalyzes ACM-B to ACM-A [37,38].

### 2.1. Gene Cluster for Biosynthesis of Aklavinone in S. galilaeus

Although anthracyclines vary in structure, their biosynthesis pathways are similar. The iterative Claisen condensation reaction is accomplished by type Ⅱ PKSs which are encoded by a gene cluster in the *Streptomyces* genome. In this cluster, the polyketide assembly requires a minimal set of iteratively used enzymes. This minimal PKS (min PKS) consists of a ketosynthase (KS_α_), a chain length factor (CLF or KS_β_), and an acyl carrier protein (ACP) [39]. In general, KS_α_ shows high sequence similarity to KS_β_ [40]. However, KS_β_ is catalytically inactive due to the lack of active-site cysteine. Instead, KS_β_ carries a highly conserved glutamine [41]. The malonyl-CoA: ACP acyltransferase (MAT) activity is usually crucial for PKS [42]. The biosynthesis of polyketide starts with iterative action of the min PKS complex to form poly-β-ketone intermediates, which are subsequently folded into distinct aromatic compounds by ketoreductases (KR), cyclases (CYC), and aromatases (ARO) [43].

Anthracyclines are assembled by tetracyclic aglycones, 7,8,9,10-tetrahydro-5,12-naphthacene-quinone, and various L-deoxysugar moieties. Aklavinone is the most common aglycones for the biosynthesis of anthracyclines. Sequence analysis reveals that a total of 12 genes participate in the biosynthesis of aklavinone in *S. galilaeus* ATCC 31615 (Table 1). Briefly, one molecule of propionyl-CoA and nine molecules of malonyl-CoAs are catalyzed into one molecule of 21-carbon decaketide nascent chain (**15**) by min PKS which contains AknB (KS_α_, EC:2.3.1.-), AknC (KS_β_, EC:2.3.1.-), AknD (ACP), AknE2 (EC:4.2.1.-), and AknF (malonyl-CoA: ACP transacylase, MAT) [33]. The growing polyketide chain accepts one 2-carbon unit from one molecule of malonyl-CoA. The interactions between AknD, AknE2, and AknF are essential for the choosing of the starter unit [44]. Only when all aforementioned enzymes are available is propionyl-CoA then chosen as the starter unit. AknA (EC1.1.1-) and AknE1(EC:4.2.1.-) are NADPH-dependent reductase and aromatase, respectively, which jointly convert the linear skeleton to 12-deoxyaklanoate (**17**) [45]. The cyclization of the second and third ring is catalyzed by AknW, leading to the formation of 12-deoxyalkalonic acid (**18**) [33,46]. This intermediate is subsequently converted to aklanonate (**19**) by mono-oyxgenase AknX-mediated oxidation. This mono-oyxgenase AknX (EC1.13.12.22) was successfully purified and biochemically verified [47]. Native AknX was suggested to manifest homotrimeric subunit structure [47]. Aklanonate (**19**) is then catalyzed into methyl aklanonate (**20**) by *aknG*-encoded (formerly *acmA*) acid methyltransferase (EC2.1.1.288) [48,49]. AknG is a SAM-dependent O-methyltransferase capable of transferring a methyl group from the donor to receptor using the ubiquitous methyl donor SAM as the cosubstrate. Aklaviketone (**21**) is formed by the closure of the fourth ring, which is catalyzed by cyclase AknH (EC5.5.1.23) [50,51]. This process is designated as intramolecular aldol condensation [52]. Lastly, reductase AknU (EC1.1.1.362) converts the 7-oxo moiety of aklaviketone into a hydroxyl group to form aklavinone (**22**) [48,53,54].

The structure analysis of PKSs not only reveals substrate specificity and protein interaction but also provides insights for combinatorial biosynthesis of derivatives based on tailored genes. However, it is challenging to determine crystal structures [44]. Of the PKSs participating in ACM-A biosynthesis, only AknH has been investigated for its structure [50]. AknH shares 61.91% identity with SnoaL, which catalyzes the ring closure in the biosynthesis of nogalamycin [52]. In addition, AknH (PDB: 2F98) forms a tetramer with 222 symmetry, like SnoaL (PDB: 1SJW) (Figure 4A). Both AknH and SnoaL utilize the same substrate due to highly similar crystal structures, but their catalytic products are different due to the difference in stereoselectivity (Figure 4B). The stereoselectivity of AknH is mainly ascribed to amino acid residues Tyr15 and Asn51, whose counterparts in SnoaL are phenylalanine and leucine [50].

### 2.2. Biosynthesis of TDP-Sugars

In the biosynthesis of antibiotics, sugar donors are tethered to aglycones, which are catalyzed by glycosyltransferases (GTs). This process is crucial for the pharmacological and pharmacokinetic properties of medicines [55,56]. By the introduction of amino groups, the modified sugars provide hydrogen bonds to the functional groups of protein or DNA targets [57]. Dedicated glycosylation leads to the formation of products with diverse recognition elements, and sugar conjugation usually improves the solubility, chemical stability, and polarity of aglycone [57].

The sugar moieties biosynthesis of ACM-A in *S. galilaeus* ATCC 31615 is mainly executed by a gene cluster consisting of *aknY*, *aknR*, *aknN*, *aknQ*, *aknP*, *aknZ*, *aknL*, *aknM*, and *aknX2* (Figure 3A) [58]. In other *Streptomyces* species, the biosynthesis of dTDP-deoxyfucose (**11**) and dTDP-rhodinose (**14**) of ACM-A is accomplished by *dnm* or *spn* clusters (Table 2) [59,60]. For sugar moieties biosynthesis in *S*. *galilaeus*, thymidylyltransferase converts glucose-1-phosphate (**1**) to TDP-D-glucose (**2**), which is subsequently converted to TDP-4-keto-6-deoxy-D-glucose (**3**) by TDP-D-glucose 4,6-dehydratase. The two steps are likely catalyzed by AknY (EC2.7.7.24) and AknR (EC4.2.1.46), respectively. AknN (EC4.2.1-) is a 2,3-dehydratase catalyzing the corresponding C-2 deoxygenation into TDP-3,4-diketo-2,6-dideoxy-D-glucose (**4**). AknZ is a putative 3-aminotransferase which transfers the amino group to the C-3 position, thereby generating TDP-3-amino-4-keto-2,3,6-trideoxy-D-glucose (**5**). AknL is proposed to show dTDP-4-dehydrorhamnose 3,5-epimerase activity and catalyze TDP-3-amino-4-keto-2,3,6-trideoxy-D-glucose into TDP-3-amino-4-keto-6-deoxy-L-glucose (**6**). AknM (EC4.2.1.164) may serve as an *aknM*-encoded 4-ketoreductase catalyzing C-4 ketone into TDP-L-daunosamine (**7**). AknX2 is an N-methyltransferase which transfers the methyl group at C-3 amine to TDP-rhodosamine (**8**) [61]. AknQ (EC1.1.1.384) is a putative 3-ketoreductase catalyzing the C-2 deoxygenation of TDP-3,4-diketo-2,6-dideoxy-D-glucose and thereby the generation of TDP-4-keto-2,6-dideoxy-D-glucose (**9**), which is then converted into TDP-4-keto-2,3,6-trideoxy-D-glucose (**12**) by 3-dehydratase AknP (EC4.2.1.164) [60]. d-TDP-rhodinose (**14**) is presumably synthesized from TDP-4-keto-2,3,6-trideoxy-D-glucose through two steps catalyzed by dTDP-4-dehydrorhamnose 3,5-epimerase and 4-keto reductase. TDP-4-keto-2,6-dideoxy-D-glucose is probably converted into dTDP-4-keto-2,6-dideoxy-beta-L-galactose by 3,5-epimerase. Unfortunately, the genes encoding these enzymes in *S. galilaeus* have not been completely elucidated.

### 2.3. Transfer of TDP-Sugars

GTs transfer glycosylated moieties from activated sugar donors to receptors. Such enzymes effectively mediate the formation of glycosidic linkage of natural oligosaccharides, glycoconjugates, and their analogues in a stereospecific and regiospecific way [62]. In view of sequence similarity and conserved motifs, the reported GTs can be classified into a total of 114 families which are deposited in the Carbohydrate-Active enzyme database (CAZy) (http://www.cazy.org/, accessed on 22 October 2021). Of these GTs, the GT1 family is most common due to its excellent glycosylation capacity [63].

In view of structures, the reported GTs can be divided into three superfamilies: GT-A, GT-B, and GT-C folds [64]. In general, the GT1 family exhibits GT-B fold. GT-B glycosyltransferases harbor two distinct β/α/β Rossmann domains which are flexibly connected [65]. The active site generally locates in the cleft between two domains [66]. The C-terminal donor binding motif is relatively conserved, whereas the N-terminal domain shows high divergence and considerable topological plasticity [67]. The binding sites mainly consist of hydrophobic amino acid residues, while the gateway to ligand entry and departure is generally rich in charged amino acid residues [68,69]. In glycosylation, GT-B proteins generally alter their conformation and shape of binding pocket when binding to the sugar donor, leading to a slight rotation of the N-terminal domain toward the C-terminal domain [70].

GTs can be divided into two types: inverting GT and retaining GT, depending on whether the configuration of the transferred sugars in end-product is inverted or retained (Figure 5A) [64,71]. It was reported that the GT1 family undergoes an inversion of glycosylation and demonstrates a direct-displacement S_N_2-like mechanism: the acceptor nucleophilic hydroxyl attacks the anomeric carbon of the sugar donor and displaces the leaving nucleotide portion on the opposite face [64,72,73]. Compared with standard glycosylation, the inverting GTs catalyze the formation of products with reversal anomeric configuration (Figure 4B).

### 2.4. Transfer of TDP-β-L-Rhodosamine

Once the polyketide stage is completed, the skeleton at the hydroxy group is glycosylated by specific GTs. In the presence of its auxiliary protein partner AknT, AknS (EC2.4.1.326) is an aklavinone 7-β-L-rhodosaminyltransferase which attaches TDP-β-L-rhodosamine to the C7-OH of aklavinone, resulting in the formation of rhodosaminyl-aklavinone (ACM-T). AknS promiscuously accepts diverse sugar donors and receptors. Consistent with this attribute, the recombinant *S. venezuelae* produces a range of DXR analogues which contain diverse deoxysugar moieties and AknS/AknT in complex with other enzymes [74]. AknS/AknT can also efficiently transfer TDP-L-daunosamine to exogenous DXR aglycone ε-rhodomycinone [61]. The above studies indicate the wide substrate scope of AknS, which provides insights for the combinatorial biosynthesis of unnatural antibiotics. It is challenging to visualize the transient interaction between AknS and AknT, which jointly form a complex, and their crystallization as well as isolation methods remain poorly understood [75]. Fortunately, AknS belongs to the GT1 family, whose comparable architectures allow for the characterization of congeneric proteins. Of the activator-dependent GTs, SpnP and EryCIII have been investigated for their crystal structures [76,77]. The above structural studies have uncovered key residues for their reactions, such as residue H13 deprotonating the corresponding hydroxy group in aglycone substrates, and residues D356 and E357 contributing to the formation of a two-residue motif (D/E-Q) which mediates the binding of sugar moieties [78]. Interestingly, a three-helix motif was recognized as a common feature and used as an identification tag of GTs that require auxiliary proteins [76]. Moreover, two amino acids (N230 and T335) were found to be crucial for the formation of a hydrogen bond with a TDP unit [76]. AlphaFold is a protein structure prediction tool with high accuracy [79], and the 3D structure of an AknS monomer predicted by the AlphaFold Monomer v2.0. Homologous analysis with other GTs reveals that AknS is arguably a homodimer. This AknS monomer contains a parallel β-sheet surrounded by α-helices, which is consistent with GT-B proteins (Figure 6A). The key residues H13 and D356 are labeled. The three-helix motif is found in the AknS sequence and comprises three helices (Figure 6B).

P450-like enzymes are fundamental for the functions of GTs. Located in the upstream of *aknS*, the gene *aknT* encodes an enzyme (AknT) which exhibits oxidoreductase activity. AknS exhibits low activity in the absence of AknT. When binding to AknS at a molar ratio of 3:1, AknT, as a regulator subunit, may optimize the fitness and productive orientation of akalvinone and thereby increase the catalytic constant of AknS by 40-fold [34]. A BLAST search using AknT sequence as a query revealed a plethora of homologous proteins which are putative P450-derived enzymes from *Streptomyces*. The predicted structure of AknT is shown in Figure 7A. AknT shows a typical cytochrome P450 fold and the conserved amino acid sequence is only limited to the C-terminal [77,80]. Notably, AknT lacks the conserved cysteine binding to heme group, indicating its difference relative to the authentic cytochrome P450 family. Such a difference presumably makes the conformation of AknT more dynamic than typical P450 [77]. Indeed, AknT transiently forms a ternary compound with AknS and substrate, which stimulates the conformational change in AknS and facilitates the transfer of sugar moieties [81]. A previous study indicated that the complex of EryCIII with its auxiliary protein EryCII manifests an α2β2 heterodimer, which is an elongated quaternary organization when the auxiliary protein resides in the periphery of the GT homodimer (Figure 7B) [77]. In this complex, EryCII may serve as a scaffold to provide an α-helix for linking EryCIII [77]. These GT-auxiliary protein pairs may assemble in the same way. It is thus speculated that the interaction surface of the AknS/AknT heteromer is different from that of the AknS homomer.

### 2.5. Transfer of 2-Deoxy-β-L-Fucose and L-Rhodinose

There exist trisaccharide chains adhered to the skeleton of ACM-A. However, only AknS and AknK have been identified from the ACM-A-producing strain, meaning that one of the two GTs (AknS and AknK) accomplishes two glycosylation reactions [34,35,81]. LanGT1 and LanGT4 are the first discovered iterative GTs [82]. AknK (EC2.4.1.327) is an L-2-deoxyfucosyltransferase transferring two L-2-deoxysugars to the axial 4-OH of anthracycline monosaccharides. In the first reaction, AknK transfers 2-deoxy-β-L-fucose from the activated donor dTDP-2-deoxy-β-L-fucose to mono-glycosylated ACM-T, leading to the generation of di-glycosylated ACM-S [35,81]. L-rhodinose is considered a terminal sugar initially added to ACM-T. Next, L-rhodinose is converted to L-cinerulose (a 2,3,6-trideoxy-4-keto-L-hexose) to form ACM-N [35]. AknK shows lower activity in catalyzing the second transfer reaction compared to the first, indicating the specificity of AknK to dTDP-rhodinose.

AknK exhibits substrate promiscuity and accepts various TDP-sugars as donors and different aglycones as receptors. In addition, its catalytic activity varies in different reactions [35]. For instance, AknK shows the highest activity in the conversion of TDP-L-daunosamine to ACM-T, followed by the conversion of TDP-l-2-deoxyfocus. Most GTs function in a regio- and stereospecific manner. SpnG is a spinosyn rhamnosyltransferase which only accepts TDP substituent in an axial position to catalyze the nucleophilic displacement reaction [78]. AknK requires a receptor with an axial hydroxyl at the nucleophilic attacking site [35].

The TDP-binding active pockets are usually conserved, and the structural specificity of GTs relies on a receptor-binding region [83]. ProteinsPlus (https://proteins.plus, accessed on 3 October 2022) can detect binding pockets for further protein–ligand interaction analyses [84,85]. Based on the structure information given by the AlphaFold Monomer v2.0 pipeline, a binding pocket was recognized (Figure 8A). A docking analysis of AknK was carried out to elucidate the binding pockets of sugar donors by building a grid covering the deep pocket consisting of the aforementioned key residues and predicted pocket. A total of 100 dockings were performed by the genetic algorithm. Results show that two binding pockets of dTDP-2-deoxyfucose and dTDP-rhodinose are located in the cleft between the N-terminal and C-terminal domains. The energy of docking for dTDP-2-deoxyfucose is 4.74 kcal/mol, but that for dTDP-rhodinose is only 2.27 kcal/mol. The interaction residues in the binding pockets are labeled in Figure 8C,E.

### 2.6. Optimization of the Catalytic Activity of GTs

As a key enzyme for the biosynthesis of ACM-A, GT can be modified to show appropriate catalytic activity. However, its substrate specificity restricts applications. Although a one-pot multi-enzyme cascade system can yield natural or unnatural compounds, the natural GTs show drawbacks such as relatively low activity, cumbersome purification, and a short half-life [86,87,88]. In addition, heterologous bacterial hosts may generate inclusion bodies [89,90]. To improve the solubility of GTs, a plethora of protein fusion partners and solubility tags have been successfully utilized [89,91,92].

The GT-catalyzed glycosidic bond is formed in a high regio- and stereo-manner. That is, promiscuous GTs may fail to recognize nonnative receptors [93]. The sugar donor versatility of GTs has been investigated and improved. For example, a study examined UrdGT1b and UrdGT1c, which involve in urdamycin A biosynthesis. Mutations of amino acids led to the altered substrate selectivity of GTs [94]. Since members of the GT-B fold family are structurally modular, GT chimeras may be generated by domain swapping, leading to the alternation of sugar donor specificity, especially when parent templates are highly identical [95]. Moreover, the domain exchange of AtUGT78D2 and AtUGT78D led to the generation of fusion proteins. Three of them demonstrated expanded sugar-donor range and enhanced catalytic activity [96].

The rational modification of the GTs related to ACM-A biosynthesis is constrained by the lack of knowledge on crystal structure. To bypass this obstacle, high-throughput screening-dependent directed evolution could be implemented. For instance, a fluorescence-based high-throughput screening (HTS) together with error-prone PCR/saturation mutagenesis was conducted to modify GTs, leading to a 200–300-fold increase in catalytic activity [97]. A high-throughput colorimetric screen in conjunction with saturation mutagenesis was also conducted to generate GT variants, which showed enhanced catalytic activity [98]. In general, rational design is a prerequisite for the molecular modification of enzymes [99]. In this regard, SEARCHGTr is a web tool for GT analysis, which provides information on donor/receptor specificity and putative substrate-binding residues [100]. The increasing crystal structures of GTs enable this web tool to accurately predict substrate-binding pockets and facilitate their redesigning. Indeed, relying on docking analysis and in silico mutagenesis, the candidate key residues in active sites can be genetically modified to improve substrate specificity and enzyme activity [101,102,103]. The screening of novel enzymes from nature is an alternative to conventional chemical synthesis by which chemoselectivity, regioselectivity, and enantioselectivity are hard to control. Indeed, extensive studies on GTs and the burgeoning transcriptome data allow not only the mining of new GTs with substrate promiscuity but also the enriching of the glycosylation toolkit for diverse sugar molecules [104,105,106]. No matter which strategies are adopted, the donor/receptor promiscuity of enzymes is required for the development of versatile biocatalysts.

### 2.7. Oxidation of Terminal Sugar Residue

Aclacinomycin oxidoreductase (AknOx) was first isolated from *S. galilaeus* ATCC 31133 [107]. Later, the AknOx secreted from *S. galilaeus* ATCC 31615 was purified to determine its crystal structure [36,108]. AknOx catalyzes two steps of modifications of terminal sugar residue, namely the four-electron oxidation of rhodinose to L-aculose. The first step is the conversion of rhodinose to cinerulose A, i.e., oxidation of the hydroxyl at C4 to the keto group (EC1.1.3.45). The second step is the elimination of two hydrogen atoms, leading to a double bond between C2 and C3 (EC1.1.3.14) [36]. That is, AknOx firstly serves as an oxidase to transfer hydride from rhodinose to FAD. In the second step, AknOx still acts on the terminal sugar as a proton/hydride redox catalyst. The cells lacking AknOx accumulate various ACM-A analogues [107]. Unlike AknOx, which catalyzes the formation of cinerulose A, GcnQ is a flavin-dependent oxidoreductase catalyzing the formation of L-aculose. In the latter reaction, no intermediate cinerulose A was detected [109].

AknOx is one member of vanillyl-alcohol oxidase/*p*-cresol methylhydroxylase (VAO/PCMH) flavoprotein family, whose catalytic motif consists of two tyrosine residues [110]. The proton abstraction in the first step catalyzed by AknOx is usually executed by Tyr-493, and Tyr-421 mediates the second reaction to generate L-aculose [36]. Clearly, AknOx is an unusual flavoenzyme, as it harnesses one active site but two different sets of catalytic residues. When tyrosine is mutated into phenylalanine, AknOx non-covalently binds to flavin and shows reduced activity [111].

AknOx is an oxygen-dependent FAD-linked oxidoreductase. Cofactor reduction is implemented by the substrate. The subsequent regeneration of flavin by reacting with molecular oxygen leads to the formation of hydrogen peroxide and the reoxidation of FAD [112]. BLASTp analysis reveals the high identity of AknOx amino acid sequences, and most of them come from the *Streptomyces* species. A large proportion of these proteins were annotated as FAD-dependent oxidoreductases or dehydrogenases containing the berberine bridge enzyme (BBE) sequence motif. In such enzymes, highly conserved His and Cys residues are characteristically linked to cofactor FAD via 8α-histidylation and 6-S-cysteinylation [113]. Of the FAD-dependent enzymes, Dbv29 is a flavin mononucleotide-dependent hexose oxidase catalyzing a four-electron oxidation reaction [114]. The isoalloxazine ring of flavin is covalently linked to the side-chains of His91 and Cys151. In AknOx, the two amino acid residues are replaced by His70 and Cys130, respectively (Figure 9C) [36,114]. This double-anchor allow enzymes showing an open active site to bind to bulky substrates and maintain architecture, thereby preventing the dissociation of the cofactor [111]. In addition, covalent attachment enhances the oxidation capacity of cofactors and remarkably improves the reduction potential [112]. AknOx harbors a Rossmann fold-binding motif (GXXGXXXG) that covalently binds to the BBE motif and adenosine moiety of FAD [115]. The solution of purified enzymes containing a flavin-like cofactor was bright yellow. Conversely, when His and Cys were mutated, AknOx did not manifest bright yellow and therefore lost catalytic activity, while Dbv29 retained the original yellow color [116].

Secretory expression is important for the large-scale production of proteins. The evolved Twin-Arginine Translocation (Tat) system can translocate fully-folded and cofactor-binding proteins across the cytoplasmic membrane [117]. The endogenous Tat system in refined model organisms benefits the large-scale production of secreted proteins and enzymes [118]. Bioinformatics analysis reveals that the N-terminal of AknOx carries a Tat signal peptide. This signal peptide benefits large-scale protein secretion in heterologous hosts.

AknOx is one of the few structurally identified enzymes participating in ACM-A biosynthesis. The crystal structure of AknOx was determined by multiwavelength anomalous diffraction [108]. In addition, its crystal structure was reported by a group from the University of Turku [36]. In its crystal, each stable unit contains four subunits of AknOx that constitute two dimers, and the AknOx harbors two domains: one is the F domain binding to FAD, the other is S domain carrying the majority of residues for interplay with the substrate (Figure 9A,B). FAD interacts with binding sites through several main- and side-chain hydrogen bonds [36]. The side-chain of ACM-A inserts into the pocket of the S domain, resulting in van der Waals between the atoms of the ligand and several hydrophobic residues [36].

### 2.8. Regulatory Genes for ACM-A Biosynthesis

In the genera of *Streptomyces*, the biosynthesis of secondary metabolites usually undergoes complicated hierarchical regulation [119,120]. First, signals are usually sensed by global and/or pleiotropic regulators. Next, signals are transferred to pathway-specific regulators of corresponding compounds and, in turn, activate biosynthesis. The *Streptomyces* Antibiotic Regulatory Protein (SARP) is a specific regulator influencing the gene clusters for the biosynthesis of secondary metabolites. SARP activates the ACM-A biosynthesis gene clusters, mainly by binding to DNA recognition sequences [121]. AknI and AknO are homologous to the SARP family which regulates gene expression. The inactivation of AknO blocks the formation of ACM-A [48]. Both AknO and AknI belong to the AfsR/DnrI/RedD regulatory family, which carries several highly conserved residues essential for DNA binding to the N-terminal [122]. AntiSMASH analysis of biosynthetic gene clusters led to the identification of other regulators such as LuxR and TetR family regulators, which are adjacent to ACM-A biosynthetic genes in *S. galilaeus* [123,124]. The regulation of ACM-A biosynthesis requires in-depth study. In *Streptomyces*, two-component signal transduction systems (TCSs) serve as a primary mechanism influencing metabolite production, virulence factors, and quorum sensing (QS). The regulatory proteins in TCSs respond to the environmental cues captured by the sensor kinase in the cell membrane. Then, kinase phosphorylates itself to respond to the signal and transfers the phosphoryl group to a regulator [125]. TCSs were mainly identified from the model strain *S. coelicolor* [126]. However, the complicated regulatory network in *S. galilaeus* remains to be unraveled.

## 3. Strategies for Improving ACM-A Production

ACM-A and its analogues were originally isolated from *S. galilaeus* ATCC 31133 broth [11]. It is challenging to separate ACM-A from its analogues due to similar structures. To increase ACM-A and accordingly reduce analogues, *S. lavendofoliae* ATCC 15872 was mutated, leading to *S. lavendofoliae* DKRS, which produced 125 mg/L ACM-A [127]. Adjusting the pH value at the late stage of fermentation could also promote ACM-B conversion to ACM-A and thereby facilitate the purification of ACM-A [128]. Interestingly, the mutants of *S. galilaeus* H026 caused by the treatment of N-methyl-N′-nitro-N-nitrosoguanidine (NTG) could convert AcmB to AcmA, but could not catalyze AcmA to AcmB, which is conducive to the industrial production of AcmA [129]. In an effort to advance aklavinone biosynthesis, a group from Stanford University leveraged bimodular PKSs to produce hexaketides and octaketides [130]. Recently, by optimizing promoters, enzymes, and chassis cells, a recombinant *Streptomyces* strain was engineered, which produced 15–20 mg/L aklavinone [131]. More importantly, a total of 37 genomes were shown to harbor the gene clusters for the biosynthesis of aclacinomycin, indicating the feasibility of screening strains for the high-level production of ACM-A [132,133].

How to improve ACM-A production may also draw lessons from the biosynthesis of other antibiotics in *Streptomyces*. The biosynthesis pathway is strictly controlled by regulatory cascades mainly consisting of pleiotropic regulators, global regulators, and feedback regulation [134]. Signal-transduction-mediated pleiotropic regulation works at the beginning of antibiotics biosynthesis. A previous study identified a γ-butyrolactone (GBL) named autoregulatory factor (A-Factor) which could promote the production of streptomycin in *S. griseus* [135]. Other GBLs can be categorized based on structure differences in their side-chain. The A-Factors bind to cytoplasmic receptors (e.g., TetR family regulators) and activate gene transcription [136]. Relying on this principle, the levels of secondary metabolites in *Streptomyces* could be enhanced by the exogenous addition of A-Factor analogues or the manipulation of signal-molecule-related genes [137,138,139]. Since SARP family proteins are pathway-specific regulators, the overexpression of SARP regulator DnrI and AfsR lead to improved production of DNR and DXR [140,141].

WblA and BldA are two highly homologous global regulators affecting antibiotic biosynthesis in *Streptomyces* [142,143]. While the deletion of the negative regulator WblA led to a 70% increase in DXR production in *S. peucetius*, the heterologous expression of *bldA* resulted in a 45.7% increase in DNR production in the same strain [140,144]. The *Streptomyces* TCSs participate in the global regulation of secondary metabolisms and presumably are related to QS [145]. The discovery of QS inducers and related mechanisms suggests the essential roles of QS in metabolic regulation of *Streptomyces* [146,147,148,149]. When an endogenous QS system was combined with CRISPR interference, the resulting circuits precisely regulated gene expression and resulted in a 560% increase in rapamycin production in *S. rapamycinicus* [150]. A Gram-negative bacterial QS system was also engineered in *Streptomyces*, leading to the improved production of oxazolomycin [151]. In particular, gene expression was regulated by DNA methylation, which hampered the binding of RNA polymerases to promoters or transcription factors to their recognization sites at a pretranscriptional level in bacteria [152,153]. Unlike DNA methylation, small regulatory RNAs (sRNAs) control gene expression at posttranscriptional level [154]. It was reported that paired-termini antisense RNAs were used to inhibit DoxR and overproduce doxorubicin [155]. Compared to sRNAs, DNA methylation seems to be more attractive in strain improvement, as it is heritable. Unfortunately, so far, DNA methylation in *Streptomyces* remains poorly understood, and in-depth study is needed to determine the epigenetic variation loci affecting metabolite formation. We believe epigenome research will offer valuable insights into the high-level production of ACM-A and beyond.

The biosynthesis of antibiotics leads to feedback inhibition, which halts cell growth and constrains their further accumulation due to toxicity and the presence of pathway-specific negative regulators [156,157]. Negative feedback can be minimized by the overexpression of resistance genes [157,158], transporters [159], and efflux-pump-coding genes [160,161]. Ribosome engineering is also effective to improve cell resistance to metabolites, especially when in conjunction with conventional mutagenesis or genome shuffling [162,163,164,165,166]. Compared with genetic modification strategies for screening secondary metabolites, small-molecule perturbation seems to be more applicable to generate mutants. For example, a group identified four molecules designated as ACR2 (antibiotic remodeling compound) which were able to inhibit the enoyl reductase activity of FabI. The inhibition of FabI blocked fatty acid biosynthesis, as it catalyzes the final and rate-limiting step [167]. While this inhibition reduced the precursors toward fatty acid biosynthesis, it increased those toward secondary metabolites, especially for CoA-dependent metabolite formation. In fact, this inhibition led to the improved production of doxorubicin, baumycin, and desferrioxamine B and E [168].

Apart from aforementioned strategies, ACM-A could also be overproduced by the overexpression of rate-limiting enzymes [169] or the genes for the glycosylation and biosynthesis of sugar moieties [170]. Incorporating a heterologous metabolic pathway into host cells may lead to the increased production of desired metabolites if the host cells provide sufficient precursors and energy and show genetic tractability [171,172]. Synthetic biology provides feasible strategies to create efficient chassis cells for heterologous protein expression [173,174]. For the overproduction of ACM-A, one challenge is how to express a functional min PKS which is insoluble in some prokaryotes [175]. The solution to this challenge may be the employment of a molecular chaperon [176] or an appropriate promoter, because the former helps proper protein folding and the latter controls transcription speed. Another issue is the perturbation of the host metabolism. The sigma factor (σs) could be leveraged to stimulate the production of the desired metabolites, as σs can allocate metabolic fluxes [177,178]. Indeed, σ-based regulation has been utilized to intensify gene expression [179,180]. In recent years, orthogonal gene circuits and modular devices are considered effective to coordinate product formation and cell growth [177,181].

One notable challenge for overproducing ACM-A is how to efficiently drive the long biosynthesis pathway. Solutions to this challenge include the coupling of the ACM-A pathway to a core pathway, the timely alleviation of feedback inhibition, the adequate utilization of metabolism [182], and the augmentation of precursor supply [183,184,185,186]. In addition, cluster-free hosts can divert the precursor to the desired chemicals [187]. Although the overexpression of key enzymes is, in most cases, feasible to overproduce desired chemicals, it exerts a metabolic burden on the host cells [188]. Apart from molecular biology approaches, the timely in situ removal of metabolites is actually feasible to minimize metabolic stress and simplify downstream purification [189,190]. Recently, a tRNA from *Streptomyces* was shown to benefit antibiotic production by circumventing inefficient wobble base-pairing, which is insightful for product formation [191]. Table 3 shows the reported strategies for improving the production of ACM-A and other secondary metabolites in *Streptomyces*.

Anthracyclines are important anticancer agents; however, their cardiotoxicity limits doses in clinical practice. Hence, it is desirable to seek new anthracycline analogues with high antitumor activity and low cardiotoxicity. 11-hydroxyaclacinomycin A is an analog showing higher in vitro cytotoxicity against melanoma and leukemia compared to ACM-A. 11-hydroxyaclacinomycin A could be produced by transforming *S. galilaeus* ATCC 31133 with doxorubicin resistance genes *drrA* and *drrB*, as well as the aklavinone 11-hydroxylase-coding gene *dnrF* from the doxorubicin producer [192]. Moreover, the isomerization of the ACM-A hydroxyl region in *S. galilaeus* mutants led to the production of iso-aclacinomycins, which showed enhanced (1~5 folds) anticancer activity relative to ACM-A [193]. Overall, a lot of approaches could be developed to overproduce ACM-A and its analogues.

## 4. Conclusions

The antitumor activity of ACM-A has brought a glimmer of hope for cancer patients. This review comprehensively describes the pathways and key enzymes for the biosynthesis of ACM-A. Although the enzymes for the biosynthesis of ACM-A have not been completely unraveled, genome sequencing and bioinformatics analysis hold promise to fasten its annotation and identification. Although current low production constrains its clinical application, a set of strategies have been developed. For instance, constraint-based modeling techniques have been developed for the accurate analysis of metabolic networks [194], and cell systems can be modified at DNA, RNA, and protein levels. For DNA modification, the genome and gene cluster can be reshaped by DNA editing tools such as Multiplex Automated Genome Engineering (MAGE), Multiplexed Site-specific Genome Engineering (MSGE) [195,196], CRISPR/Cas9, base editing [197,198], and prime editing [199]. CRISPR-Cas9 holds the potential to rewire the metabolic pathways and regulatory networks in *Streptomyces* [200,201]. Transcription can be modulated by RNA interference, anti-sense technology, and CRISPR-Cas13 technology, especially in eucaryotic host cells [202]. Notably, the combination of CRISPR-dCas9 with DNA methylation seems to be a promising strategy for targeted gene knockdown [203]. Apart from the modifications to DNA and RNA, protein engineering is ushering in a new era, as structure prediction tools, especially AlphaFold, help the directed evolution of the enzymes that catalyze the formation of desired metabolites [79]. We envision that the aforementioned strategies and tools will substantially advance the bioproduction of ACM-A and beyond.

## Figures and Tables

**Figure 1 molecules-28-02761-f001:**
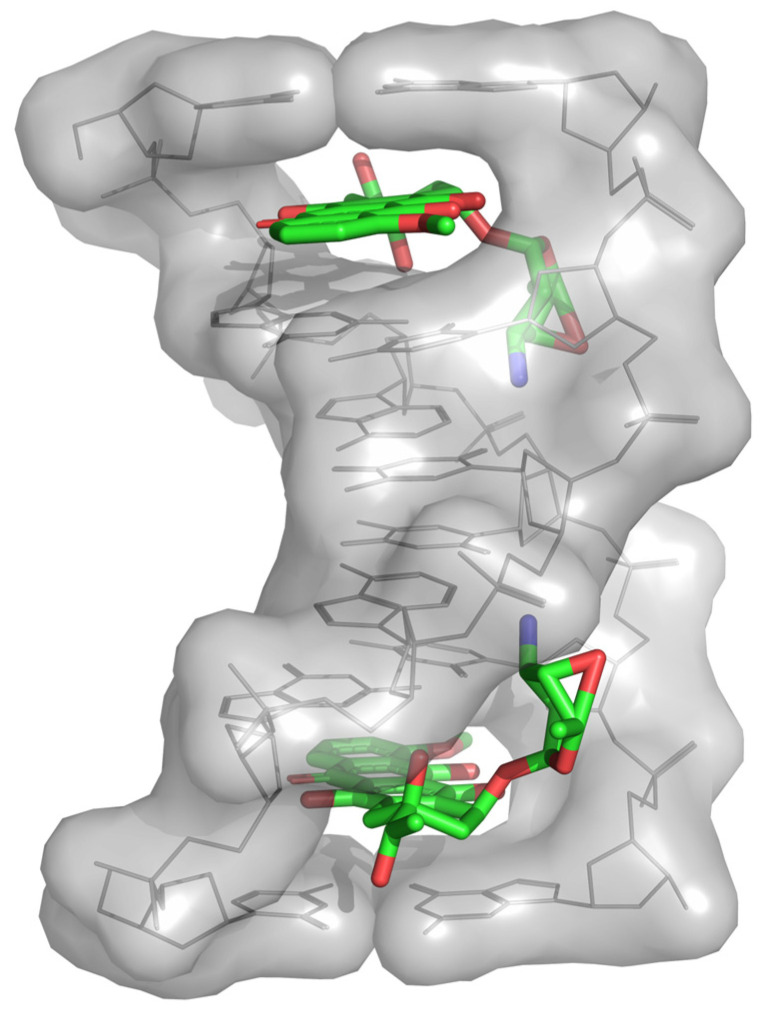
The crystal structure of DNA–daunomycin complex (PDB 1DA0).

**Figure 2 molecules-28-02761-f002:**
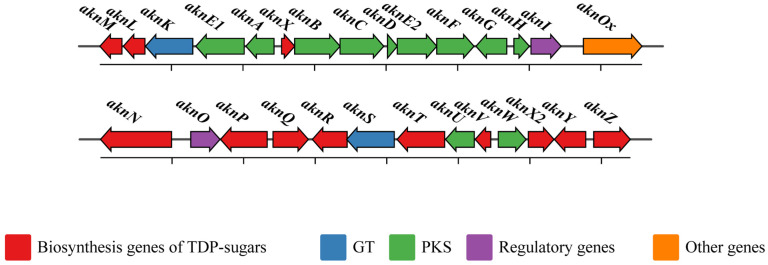
Organization of the aclacinomycins biosynthesis gene cluster. Arrows indicate the directions of gene transcription. GT, glycosyltransferase; PKS, polyketide synthase.

**Figure 3 molecules-28-02761-f003:**
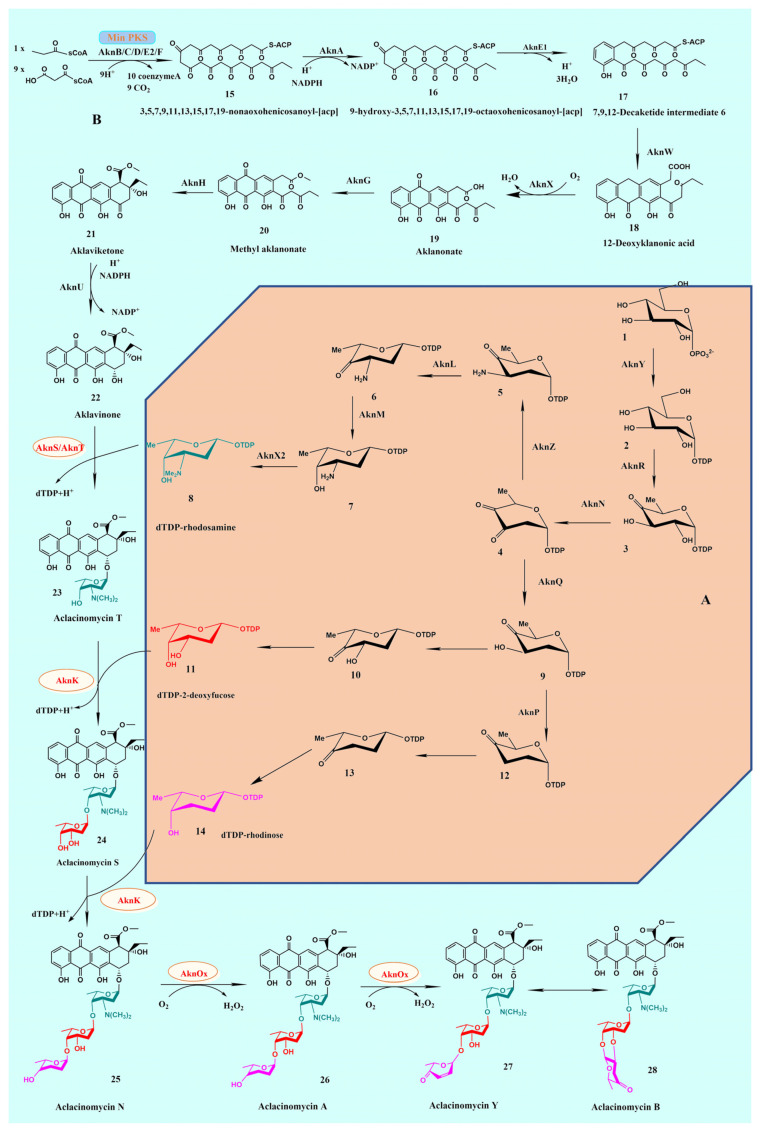
Biosynthesis pathway of aclacinomycin A. (**A**) Biosynthesis of TDP-sugars; (**B**) Type II PKS pathway and post-PKS tailoring modifications.

**Figure 4 molecules-28-02761-f004:**
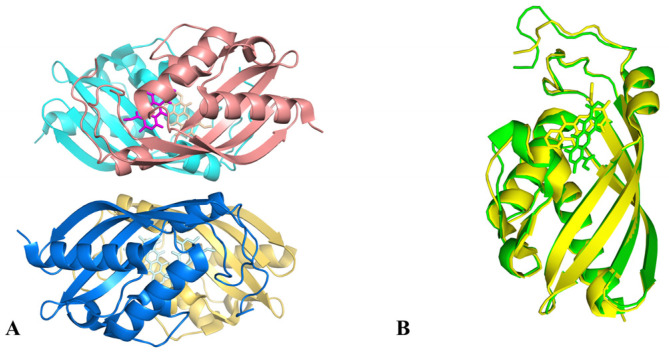
Structure information of AknH. (**A**) Structure of AknH; (**B**) Superimposition built by Pymol. AknH and SnoaL are shown in green and yellow, respectively.

**Figure 5 molecules-28-02761-f005:**
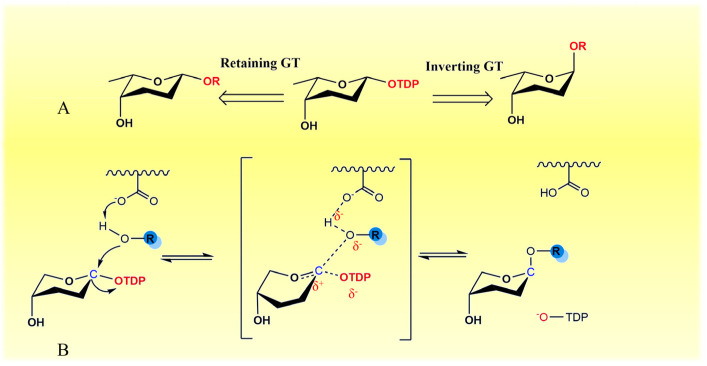
Catalytic mechanisms of glycosyltransferases. (**A**) Reaction mechanism of inverting and retaining glycosyltransferases; (**B**) Inverting glycosyltransferase by a direct-displacement S_N_2-like reaction mechanism.

**Figure 6 molecules-28-02761-f006:**
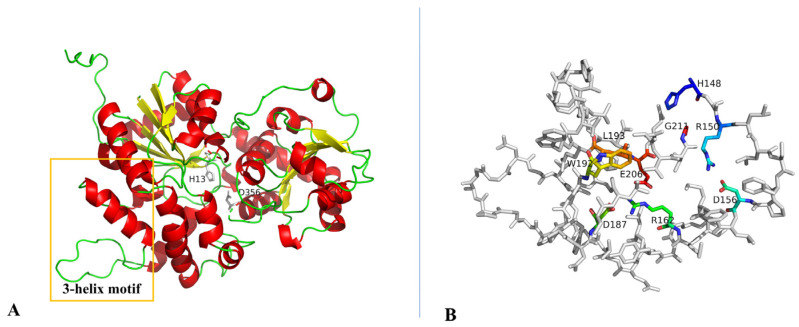
Predicted structure module of AknS. (**A**) AknS monomer prediction carried by AlphaFold Monomer v2.0; (**B**) The three-helix motif in AknS with labels of highly conserved residues.

**Figure 7 molecules-28-02761-f007:**
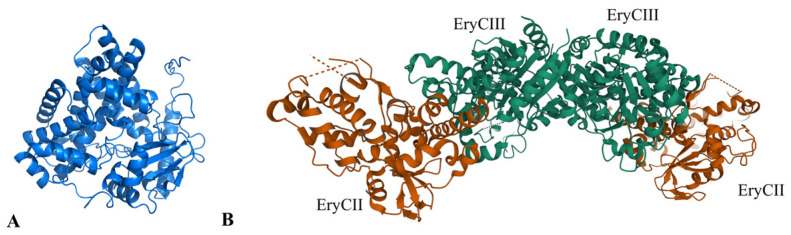
Structure information of AknT. (**A**) Predicted structure model of AknT; (**B**) The cartoon image of ErycIII in complex of EryCII.

**Figure 8 molecules-28-02761-f008:**
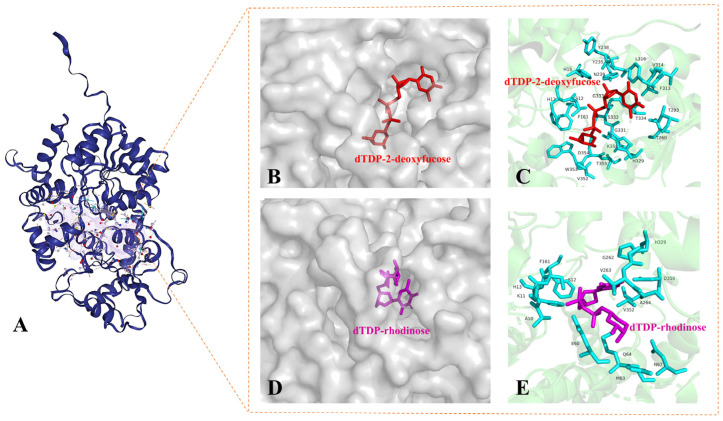
Homology model of AknK and docking analysis. (**A**) Homology model of AknK; the close-up view of the profile maps showing the active site binding as well as enlarged views of molecular docking of AknK with (**B**,**C**) dTDP-2-deoxyfucose; and (**D**,**E**) dTDP-rhodinose, respectively.

**Figure 9 molecules-28-02761-f009:**
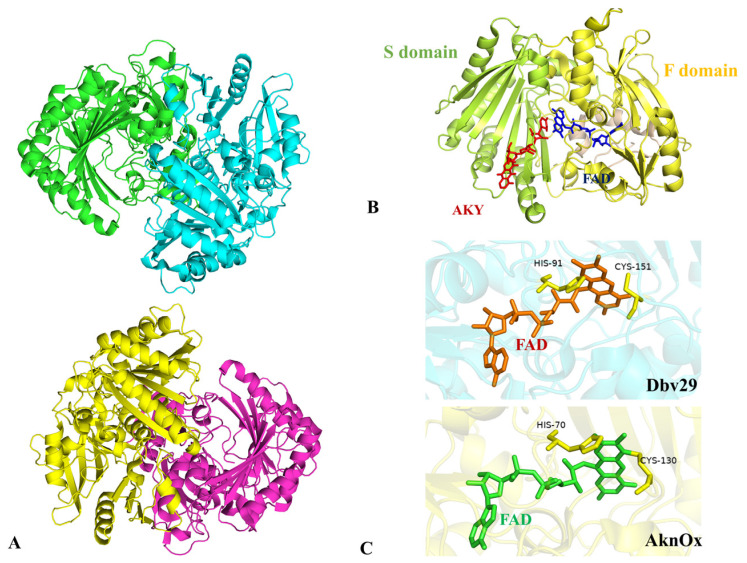
Structure information of AknOx. (**A**) Structure of AknOx; (**B**) The S domain and F domain are shown in yellow and light green, respectively; (**C**) FAD-binding pocket of AknOx and Dbv29.

**Table 1 molecules-28-02761-t001:** The identified and putative functions of the *akn* genes.

Gene Product	Number of Amino Acid Residues	Uniprot	Functions
AknA	261	Q9L553	β-ketoreductase
AknB	423	Q9L551	KS_α_
AknC	407	Q9L550	KS_β_
AknD	91	Q9L549	Acyl carrier protein (ACP)
AknE1	450	Q9L554	Aromatase (ARO)
AknE2	368	Q9L548	Determines the starter unit
AknF	347	Q9L547	Malonyl-CoA: ACP transacylase (MAT)
AknW	259	Q9L4U2	Cyclase
AknX	122	Q9L552	Mono-oxygenase (OXY)
AknG	286	Q9L546	Methyl transferase (MET)
AknH	144	O52646	Aklanonic acid methyl ester cyclase (AAME-cyclase)
AknU	267	Q9L4U4	Aklaviketone reductase (KRII)

**Table 2 molecules-28-02761-t002:** The identified and putative functions of akn enzymes and isozymes.

Gene Product	Number of Amino Acid Residues	Uniprot	Functions	Isozyme
AknY	291	Q9L4U0	Glucose-1-phosphate thymidylyltransferase	DnmL
AknR	323	Q9L4U7	dTDP-Glucose 4,6-dehydratase	DnmM
AknN	662	Q9L4V1	2,3-Dehydratase	DnmT,
AknQ	329	Q9L4U8	3-Ketoreductase	SpnN
AknP	434	Q9L4U9	3-Dehydratase	SpnQ
AknZ	341	Q9L4T9	Aminotransferase	DnmJ,
AknL	201	Q9L556	3,5-Epimerase	DnmU,
AknM	206	Q9L557	Reductase	DnmV
AknX2	238	Q9L4U1	N-methyltransferase	–

**Table 3 molecules-28-02761-t003:** Strategies for improving the desired metabolites in *Streptomyces*.

Organism	Secondary Metabolite	Strategy	Production Improved	Ref.
*S. lavendofoliae*	ACM-A	NTG mutagenesis	300%	[127]
*S. coelicolor*	aklavinone	optimization of promoters, vectors, and chassis strains	>100%	[131]
*S. galilaeus*	ACM-A	adjusting broth pH	Data not shown	[128]
*S. galilaeus*	ACM-A	NTG mutagenesis	Data not shown	[129]
*S. coelicolor*	DXR/DNR/akavinone	disruption of global downregulator gene	30%	[144]
*S. peucetius*	daunorubicin	overexpression of resistance genes	410%	[157]
*S. peucetius*	daunorubicin	overexpression of global regulator	45.7%	[140]
*S. peucetius*	doxorubicin	overexpression of rate-limiting enzymes	86%	[169]
*S. peucetius*	doxorubicin	expression of structural sugar biosynthesis and glycosyltransferase genes	460%	[170]
*S. avermitilis*	avermectin	ribosomal engineering	50%	[178]
*S. rochei*	Lankacidin/lankamycin	deletion of GBL receptors	>300%	[138]
*S. hygroscopicus*	validamycin	exogenous addition of A-Factor analogues	30%	[137]
*S. tsukubaensis*	tacrolimus	expression of GBL synthetase	36%	[139]
*S. rimosus*	oxytetracycline	overexpression of transporters	60%	[159]
*S.* coelicolor	germicidins	exogenous addition of ARC2	Data not shown	[167]

## Data Availability

Not applicable.

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
