# Peer review of "Rethinking Biosynthesis of Aclacinomycin A"

_molecules, 2023, doi:10.3390/molecules28062761_

Round 1
Reviewer 1 Report
This manuscript provides a comprehensive review on the biosynthesis of the clinically used antitumor agent ACM-A, which is mainly produced by chemical synthesis. It would give a hint to shift the present industrial method to an ecological and economical one. The manuscript would be accepted after examining the following points.
1. To overview the biosynthetic machinery, please introduce the organization of the biosynthetic gene cluster by using a figure.
2. I think that readers wish to know the detail achievements on productivity optimization rather than only listing rich references. Although two specific production data (lines 440 and 454) are shown, I suggest that the authors show, e. g., with a Table, a comprehensive summary of the productivity of ACM-A by genetically manipulated organisms in comparison with traditional culture of wild type Streptomyces strains. The authors actually describe “a lot of approaches have been developed to overproduce ACM-A and its analogues” (line 490-491). How mach the agent was so far “overproduced” will provide readers with insight into the future sustainable production of this cancer agent.
<minor points>
3. line 83-85: The relation between A, B, Y is not very clear. How about adding an equation of A<-->B in Fig. 2.
4. line 107: The number of genes is 12 if count E1 and E2.
5. Fig.2: Revise the bond connection of OH-/NH2- to -OH/-NH2.
6. line 162-163, and other many: d-glucose must be D-glucose
7. line 169: TDP-4-keto-6-deoxy-L-glucose (6) lacks “3-amino”.
8. Some Figures: contain very small letters that can not be read (e.g., Fig. 2, 4).
9. line 207: Replace “receptor” with “acceptor”.
Author Response
Point-by-point response to comments
Dear editors and reviewers,
Thanks for your valuable comments and suggestions. We have carefully revised the manuscript, and the revised words and sentences are shown in red.
Reviewer 1
This manuscript provides a comprehensive review on the biosynthesis of the clinically used antitumor agent ACM-A, which is mainly produced by chemical synthesis. It would give a hint to shift the present industrial method to an ecological and economical one. The manuscript would be accepted after examining the following points.
- To overview the biosynthetic machinery, please introduce the organization of the biosynthetic gene cluster by using a figure.
Reply: Figure 2 is added to show the biosynthesis gene clusters of ACM-A BGC.
Fig. 2 Biosynthesis gene cluster of aclacinomycins.
- I think that readers wish to know the detail achievements on productivity optimization rather than only listing rich references. Although two specific production data (lines 440 and 454) are shown, I suggest that the authors show, e. g., with a Table, a comprehensive summary of the productivity of ACM-A by genetically manipulated organisms in comparison with traditional culture of wild type Streptomyces strains. The authors actually describe “a lot of approaches have been developed to overproduce ACM-A and its analogues” (line 490-491). How mach the agent was so far “overproduced” will provide readers with insight into the future sustainable production of this cancer agent.
Reply: We add Table 3 to show the strategies improving the production of ACM-A and other secondary metabolites in Streptomyces. The strategies are also mentioned in section 3. Strategies for improving ACM-A production.
<minor points>
- line 83-85: The relation between A, B, Y is not very clear. How about adding an equation of A<-->B in Fig. 2.
Reply: We have added it. (see Fig. 3 in revised manuscript or below).
- line 107: The number of genes is 12 if count E1 and E2.
Reply: We have corrected the number of genes.
- 2: Revise the bond connection of OH-/NH2- to -OH/-NH2.
Reply: We have revised all the bond connection in original Fig. 2 (now Fig. 3).
- line 162-163, and other many: d-glucose must be D-glucose
Reply: All “d-glucose” have been changed to “D-glucose” (see section 2.2. Biosynthesis of TDP-sugars).
- line 169: TDP-4-keto-6-deoxy-L-glucose (6) lacks “3-amino”.
Reply: “3-amino” has been included.
- Some Figures: contain very small letters that can not be read (e.g., Fig. 2, 4).
Reply: We have updated Fig. 3 and Fig. 5. Below is Fig. 5 (the original Fig. 4).
- line 207: Replace “receptor” with “acceptor”.
Reply: revised.
Apart from above revisions, we have gone through the entire manuscript to avoid grammar errors as well as the inappropriate description. Thank you again for valuable comments and suggestions.
Best regards
Pingfang Tian
On behalf of all authors

Reviewer 2 Report
The most promising approach seems to be the study of regulatory genes that influence antibiotic biosynthesis in Streptomyces spp. it helps to increase the yield of antibiotics via molecular manipulation .
In Streptomyces, the most abundant pleiotropic regulators belong to the TCSs (two-component systems ,are the most important transduction signal mechanism in bacteria, allowing the translation of these rapid environmental or nutritional changes into a regulatory readout).
Question 1: Is it valid for this antibiotic that they are the predominant signal transduction systems in bacteria.
The manipulation of regulatory cascades can be an efficient way to improve the production of this antibiotic.
Question 2: Do you have any information on this approach?
Question 3: What about Rewiring regulatory network combined with metabolic engineering to have a more powerful way to enhance the production of this antibiotic in Streptomyces.
Please complete your review by including the following points (in order to enhance production):
- - Overexpression of positive regulator, inactivation of negative regulator, tuning feedback and ribosomal engineering.
- Binding to a specific receptor, like the A factor, is proposed as possible mechanisms of activation of secondary metabolite biosynthesis..
- The perturbation of host metabolism can be another reason to activate cryptic clusters.
- The ARC2 series, similar to triclosan, repress fatty acid biosynthesis by inhibiting the enoyl reductase FabI and change the flux of precursor molecules to antibiotic biosynthesis.
- Quorum sensing molecules could are also important factors to induce the secondary metabolite production in actinomycetes.
Question 5: It would be interesting to examine the roles of the post-transcriptional regulators in the regulation of antibiotic biosynthesis.
Little is known about regulation at pretranscriptional level (epigenetic regulation) and the posttranscriptional level, via smallnoncoding RNAs (sRNAs).
The conclusion must be reconsidered to reflect the content of the text and offer easily implementable perspectives.
Author Response
Point-by-point response to comments
Dear editors and reviewers,
Thanks for your valuable comments and suggestions. We have carefully revised the manuscript, and the revised words and sentences are shown in red.
Reviewer 2
The most promising approach seems to be the study of regulatory genes that influence antibiotic biosynthesis in Streptomyces spp. it helps to increase the yield of antibiotics via molecular manipulation.
In Streptomyces, the most abundant pleiotropic regulators belong to the TCSs (two-component systems, are the most important transduction signal mechanism in bacteria, allowing the translation of these rapid environmental or nutritional changes into a regulatory readout).
Question 1: Is it valid for this antibiotic that they are the predominant signal transduction systems in bacteria.
The manipulation of regulatory cascades can be an efficient way to improve the production of this antibiotic.
Reply: Yes, TCSs play important roles in regulating the metabolisms of Streptomyces, and the functions of TCSs have been investigated especially in S. coelicolor (Sanchez de la Nieta et al., 2022). Previous studies pointed out that regulatory genes could be leveraged to enhance the production of secondary metabolites (Yu et al., 2012; Arroyo-Perez et al., 2019). However, little is known about the TCSs, hypothetical response regulators, and histidine kinases in S. galilaeus. Clearly, deciphering the regulatory network in S. galilaeus would benefit overproduction of ACM-A.
In the revised manuscript, we discuss the above points and include several references.
Sanchez de la Nieta, R.; Santamaria, R. I.; Diaz, M., Two-component systems of Streptomyces coelicolor: an intricate network to be unraveled. Int. J. Mol. Sci. 2022, 23.
Yu, Z.; Zhu, H.; Dang, F.; Zhang, W.; Qin, Z.; Yang, S.; Tan, H.; Lu, Y.; Jiang, W., Differential regulation of antibiotic biosynthesis by DraR-K, a novel two-component system in Streptomyces coelicolor. Mol. Microbiol. 2012, 85, 535-556.
Arroyo-Perez, E. E.; Gonzalez-Ceron, G.; Soberon-Chavez, G.; Georgellis, D.; Servin-Gonzalez, L., A novel two-component system, encoded by the sco5282/sco5283 genes, affects Streptomyces coelicolor morphology in liquid culture. Front Microbiol 2019, 10, 1568.
Question 2: Do you have any information on this approach?
Reply: A panel of approaches have been developed for high-level production of antibiotics, including timely removal of desired antibiotic from fermentation broth, overexpression of positive regulator genes, attenuation of repressor genes, and global regulation-based enhancement of cell tolerance to antibiotics. However, how to overproduce ACM-A remains poorly understood. Although mutagenesis is not a new method, it is feasible to improve ACM-A production (Kim et al., 1995). Manipulation of regulatory cascades is relatively challenging due to poor understanding of their influences on ACM-A biosynthesis in S. galilaeus. Nevertheless, we could be inspired by biosynthesis of other anthracyclines such as doxorubicin and daunorubicin (Malla et al., 2010; Noh et al., 2010; Pokhrel et al., 2016). Below are references.
Kim, W.-S.; Youn, D.-J.; Cho, W.-T.; Kim, M.-K.; Kim, H.-R.; Rhee, S.-K.; Choi, E.-S. Improved production, and purification of aclacinomycin-a from Streptomyces lavendofoliae DKRS. J. Microbiol. Biotechnol. 1995, 5, 297-301.
Noh, J. H.; Kim, S. H.; Lee, H. N.; Lee, S. Y.; Kim, E. S., Isolation and genetic manipulation of the antibiotic down-regulatory gene, wblA ortholog for doxorubicin-producing Streptomyces strain improvement. Appl. Microbiol. Biotechnol. 2010, 86, 1145-1153.
Malla, S.; Niraula, N. P.; Liou, K.; Sohng, J. K., Improvement in doxorubicin productivity by overexpression of regulatory genes in Streptomyces peucetius. Res. Microbiol. 2010, 161, 109-117.
Pokhrel, A. R.; Chaudhary, A. K.; Nguyen, H. T.; Dhakal, D.; Le, T. T.; Shrestha, A.; Liou, K.; Sohng, J. K., Overexpression of a pathway specific negative regulator enhances production of daunorubicin in bldA deficient Streptomyces peucetius ATCC 27952. Microbiol. Res. 2016, 192, 96-102
Question 3: What about Rewiring regulatory network combined with metabolic engineering to have a more powerful way to enhance the production of this antibiotic in Streptomyces.
Please complete your review by including the following points (in order to enhance production):
- - Overexpression of positive regulator, inactivation of negative regulator, tuning feedback and ribosomal engineering.
- Binding to a specific receptor, like the A factor, is proposed as possible mechanisms of activation of secondary metabolite biosynthesis.
- The perturbation of host metabolism can be another reason to activate cryptic clusters.
- The ARC2 series, similar to triclosan, repress fatty acid biosynthesis by inhibiting the enoyl reductase FabI and change the flux of precursor molecules to antibiotic biosynthesis.
- Quorum sensing molecules could are also important factors to induce the secondary metabolite production in actinomycetes.
Reply: In view of your suggestion, we have revised the manuscript (see section 3. Strategies for improving ACM-A production).
Question 5: It would be interesting to examine the roles of the post-transcriptional regulators in the regulation of antibiotic biosynthesis.
Little is known about regulation at pretranscriptional level (epigenetic regulation) and the posttranscriptional level, via small noncoding RNAs (sRNAs).
The conclusion must be reconsidered to reflect the content of the text and offer easily implementable perspectives.
Reply: DNA methylation is an typical epigenetic regulation and a research hotspot especially when combined with CRISPR-dCas9. (Vojta A, Dobrinić P, Tadić V, Bočkor L, Korać P, Julg B, Klasić M, Zoldoš V. Repurposing the CRISPR-Cas9 system for targeted DNA methylation. Nucleic Acids Res. 2016, 44(12): 5615-5628. ).
Gene expression can be regulated by DNA methylation, for example, it can modulate the binding of RNA polymerase to promoters or transcription factors to recognization sites at pretranscription level in bacteria [152, 153].
Unlike DNA methylation which regulates gene expression at pretranscriptional level, small noncoding RNAs (sRNAs) regulate gene expression at potstranscriptional level [154]. To improve doxorubicin production, a paired-termini antisense RNAs were employed to inhibit DoxR [155].
So far, DNA methylation in Streptomyces remains poorly understood, and in-depth study is needed to determine the epigenetic variation loci affecting metabolites formation. We believe epigenome research will greatly advance bioproduction of ACM-A and beyond.
We have added sentences in revised manuscript (see section 3. Strategies for improving ACM-A production). Below are the references.
- Adhikari, S.;Curtis, P. D. DNA methyltransferases and epigenetic regulation in bacteria. FEMS Microbiol. Rev. 2016,40, 575-591.
- Ghosh, D.; Veeraraghavan, B.; Elangovan, R.; Vivekanandan, P. Antibiotic resistance and epigenetics: more to it than meets the eye. Antimicrob. Agents Chemother. 2020,64.
- Na, D.; Yoo, S. M.; Chung, H.; Park, H.; Park, J. H.;Lee, S. Y. Metabolic engineering of Escherichia coliusing synthetic small regulatory RNAs. Nat. Biotechnol. 2013,31, 170-174.
- Chaudhary, A. K.; Pokhrel, A. R.; Hue, N. T.; Yoo, J. C.; Sohng, J. K. Paired-termini antisense RNA mediated inhibition of DoxR in Streptomyces peucetiusATCC 27952. Biotechnol. Bioprocess Eng. 2015,20, 381-388.
- Vojta, A.; Dobrinić, P.; Tadić, V.; Bočkor, L.; Korać, P.; Julg, B.; Klasić, M.; Zoldoš, V. Repurposing the CRISPR-Cas9 system for targeted DNA methylation. Nucleic Acids Res. 2016, 44(12), 5615-5628.
We have also revised the conclusion section.
Apart from above revisions, we have gone through the entire manuscript to avoid grammar errors as well as the inappropriate description. Thank you again for valuable comments and suggestions.
Best regards
Pingfang Tian
On behalf of all authors

Round 2
Reviewer 1 Report
Thank you for the efforts for revisions. As an additional minor point, please enlarge the gene names in Figure 2 and add a brief caption about the meaning for the gene colours.
Reviewer 2 Report
The topic of the manuscript is now appropriately suited for publication in this Journal . The manuscript is well re-written and organized.
